# Photoreceptor disc incisures form as an adaptive mechanism ensuring the completion of disc enclosure

Tylor R Lewis[1]*, Sebastien Phan[2], Carson M Castillo[1], Keun-Young Kim[2], Kelsey Coppenrath[3], William Thomas[3†], Ying Hao[1], Nikolai P Skiba[1], Marko E Horb[3], Mark H Ellisman[2], Vadim Y Arshavsky[1,4]*

[1]Department of Ophthalmology, Duke University Medical Center, Durham, United States; [2]National Center for Microscopy and Imaging Research, School of Medicine, University of California, San Diego, La Jolla, United States; [3]Eugene Bell Center for Regenerative Biology and Tissue Engineering and National Xenopus Resource, Woods Hole, United States; [4]Department of Pharmacology and Cancer Biology, Duke University Medical Center, Durham, United States

*For correspondence:
tylor.lewis@duke.edu (TRL);
vadim.arshavsky@duke.edu
(VYA)

Present address: †Department of Ecology and Evolution, Stony Brook University, Stony Brook, New York, United States

**Competing interest:** The authors declare that no competing interests exist.

**Abstract** The first steps of vision take place within a stack of tightly packed disc-shaped membranes, or 'discs', located in the outer segment compartment of photoreceptor cells. In rod photoreceptors, discs are enclosed inside the outer segment and contain deep indentations in their rims called 'incisures'. The presence of incisures has been documented in a variety of species, yet their role remains elusive. In this study, we combined traditional electron microscopy with three-dimensional electron tomography to demonstrate that incisures are formed only after discs become completely enclosed. We also observed that, at the earliest stage of their formation, discs are not round as typically depicted but rather are highly irregular in shape and resemble expanding lamellipodia. Using genetically manipulated mice and frogs and measuring outer segment protein abundances by quantitative mass spectrometry, we further found that incisure size is determined by the molar ratio between peripherin-2, a disc rim protein critical for the process of disc enclosure, and rhodopsin, the major structural component of disc membranes. While a high perpherin-2 to rhodopsin ratio causes an increase in incisure size and structural complexity, a low ratio precludes incisure formation. Based on these data, we propose a model whereby normal rods express a modest excess of peripherin-2 over the amount required for complete disc enclosure in order to ensure that this important step of disc formation is accomplished. Once the disc is enclosed, the excess peripherin-2 incorporates into the rim to form an incisure.

## Editor's evaluation

This study describes the importance of the stoichiometric relationship between rhodopsin and peripherin-2 in mouse rod outer segments. The authors use genetic manipulations to vary the ratio of these two proteins to demonstrate that excess peripherin leads to excess perimeter, which then leads to the infolded structures known as incisures. These data illustrate a fundamental principle that relates factors that control the area of a structure to factors that control the perimeter of a structure.

## Introduction

In the vertebrate retina, photoreceptor cells perform the first step of vision by capturing light and generating a neuronal signal. This process, called phototransduction, occurs within a stack of

**Figure 1.** Schematic illustration of incisure arrangements. (**A**) Cartoon illustrating the structure of rod photoreceptors in mice and frogs. In each species, the outer segment contains hundreds of disc membranes in a stack. For simplicity, the connecting cilium and the axoneme are not shown. Of note, rod outer segments in frogs are much wider than in mice. (**B**) Cartoon illustrating two different types of incisure arrangements in a stack of rod discs. Incisures are indentations of the disc rim that are longitudinally aligned across the disc stack. Mouse discs have a single incisure, whereas frog discs have multiple incisures.

disc-shaped membranes, or 'discs', confined within the ciliary outer segment (see *Figure 1A* for a schematic representation). The outer segment undergoes continuous renewal whereby new discs are added at its base and old discs are phagocytosed from its tip by the apposing retinal pigment epithelium. New discs are formed as serial evaginations of the ciliary membrane, which eventually separate from the outer segment plasma membrane. This separation, known as disc enclosure, can be either complete in rods or partial in mammalian cones. The resulting mature disc contains a highly curved 'rim' supported by large oligomeric complexes of two tetraspanin proteins, peripherin-2 and ROM1 (see *Spencer et al., 2020* for a recent review on disc morphogenesis).

Early ultrastructural studies revealed that fully enclosed discs contain longitudinally aligned indentations of their rims, termed 'incisures' (*Sjostrand, 1953*; *Cohen, 1960*; *Nilsson, 1965*). Depending on the species, the number of incisures varies greatly: while mouse rods have only a single incisure (*Cohen, 1960*), frog rods have up to 20 (*Nilsson, 1965*). These incisure arrangements are illustrated in *Figure 1B*. Considering their functional role, it has been proposed that incisures can promote longitudinal diffusion of signaling molecules in the outer segment during the course of a light response (*Ichikawa, 1996*; *Holcman and Korenbrot, 2004*; *Caruso et al., 2006*; *Bisegna et al., 2008*; *Gross et al., 2012*; *Makino et al., 2012*; *Caruso et al., 2020*).

Little is known about the cellular and molecular mechanisms underlying the formation of incisures, including the relationship between the processes of disc enclosure and incisure formation. Work by Steinberg and colleagues (*Steinberg et al., 1980*) has suggested that incisure formation is not dependent on disc enclosure, based on the occasional observation of small discontinuities in the newly forming 'open' discs at the base of the outer segment. Yet, these discontinuities were not

aligned longitudinally, suggesting that they may not have been incisures. In this study, we investigated the spatiotemporal relationship between disc enclosure and incisure formation by combining traditional transmission electron microscopy (TEM) with 3D electron tomography, which provides a high-resolution reconstruction of the entire outer segment base where new discs are formed. This analysis revealed that incisures are formed immediately after discs are fully enclosed. Additionally, we observed that, as new discs evaginate from the ciliary membrane, they are not round as typically depicted but rather are highly irregular in shape and resemble expanding lamellipodia of motile cells.

Considering the molecular mechanism of incisure formation, it has been shown that variations in the expression level of rhodopsin, the predominant protein constituent of disc membranes, can influence incisure size. While incisures of WT mice extend approximately halfway into the disc, the incisures of rhodopsin hemizygous mice extend nearly the entire disc diameter (*Makino et al., 2012*; *Price et al., 2012*). In contrast, transgenic overexpression of rhodopsin reduces incisure length or precludes incisure formation (*Wen et al., 2009*; *Price et al., 2012*). Another parameter controlled by rhodopsin expression level is disc diameter, which is decreased by a reduction in rhodopsin expression (*Liang et al., 2004*; *Makino et al., 2012*; *Price et al., 2012*) and increased by rhodopsin overexpression (*Wen et al., 2009*; *Price et al., 2012*). Together, these observations led Makino and colleagues to hypothesize that each disc has a set number of molecules responsible for the formation of its rim; smaller discs containing less rhodopsin direct more of these molecules to incisures, whereas larger discs require them to be incorporated along their circumference (*Makino et al., 2012*).

The two proteins most critical for disc rim formation are the homologous tetraspanins, peripherin-2 and ROM1, which form very large oligomeric structures fortifying the entire disc rim (*Stuck et al., 2016*; *Pöge et al., 2021*). The knockout of peripherin-2 (but not ROM1) completely precludes disc formation (*Cohen, 1983*; *Jansen and Sanyal, 1984*; *Clarke et al., 2000*), while transgenic overexpression of some peripherin-2 mutants in frogs can disrupt incisure structure (*Tam et al., 2004*; *Milstein et al., 2020*). This makes peripherin-2 the prime candidate for being the limiting disc rim component postulated in *Makino et al., 2012*. Therefore, we explored the role of peripherin-2 in incisure formation in the current study. We analyzed outer segments of genetically modified mice and frogs, in which the relative outer segment content of peripherin-2 was either reduced or increased. We found that a

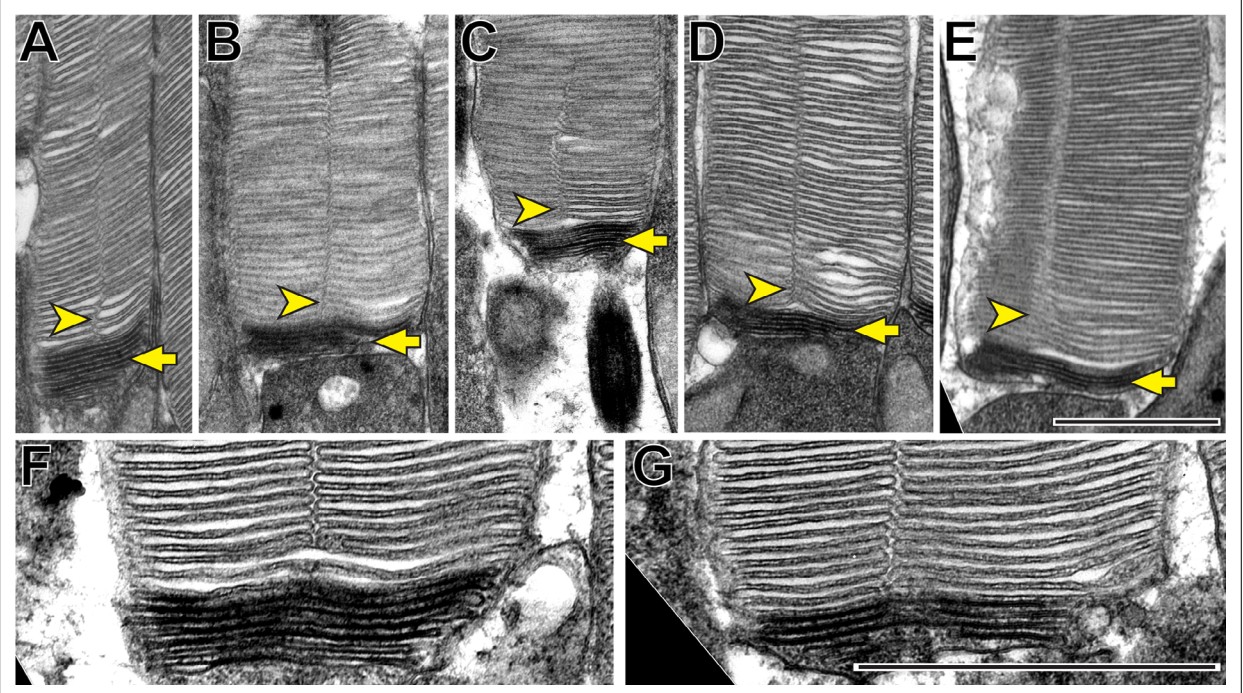

**Figure 2.** Incisures are observed only in fully enclosed discs of mouse rods. (**A–G**) Representative TEM images of longitudinally sectioned WT mouse rods contrasted with a combination of tannic acid and uranyl acetate, which stains newly forming 'open' discs more intensely than mature enclosed discs. Yellow arrows point to darkly stained, unenclosed discs; yellow arrowheads point to longitudinally aligned incisures. Scale bars: 1 μm.

high level of peripherin-2 causes an increase in incisure size and structural complexity, while a low level precludes incisure formation.

Taken together, our findings suggest that rods express a modest excess of peripherin-2 over the amount needed for completing disc enclosure. Once enclosure is finished, an excess pool of peripherin-2 remaining in the disc incorporates into the rim to form an incisure.

## Results

### Incisure formation follows the completion of disc enclosure

We first sought to determine the spatiotemporal relationship between incisure formation and enclosure in newly forming discs. Our initial approach was to use traditional TEM to image longitudinally sectioned mouse rod outer segments that had been contrasted with a combination of tannic acid and uranyl acetate. This contrasting technique (*Ding et al., 2015*) allows for the discrimination between newly forming 'open' discs that are stained darker than fully enclosed discs. Several examples shown

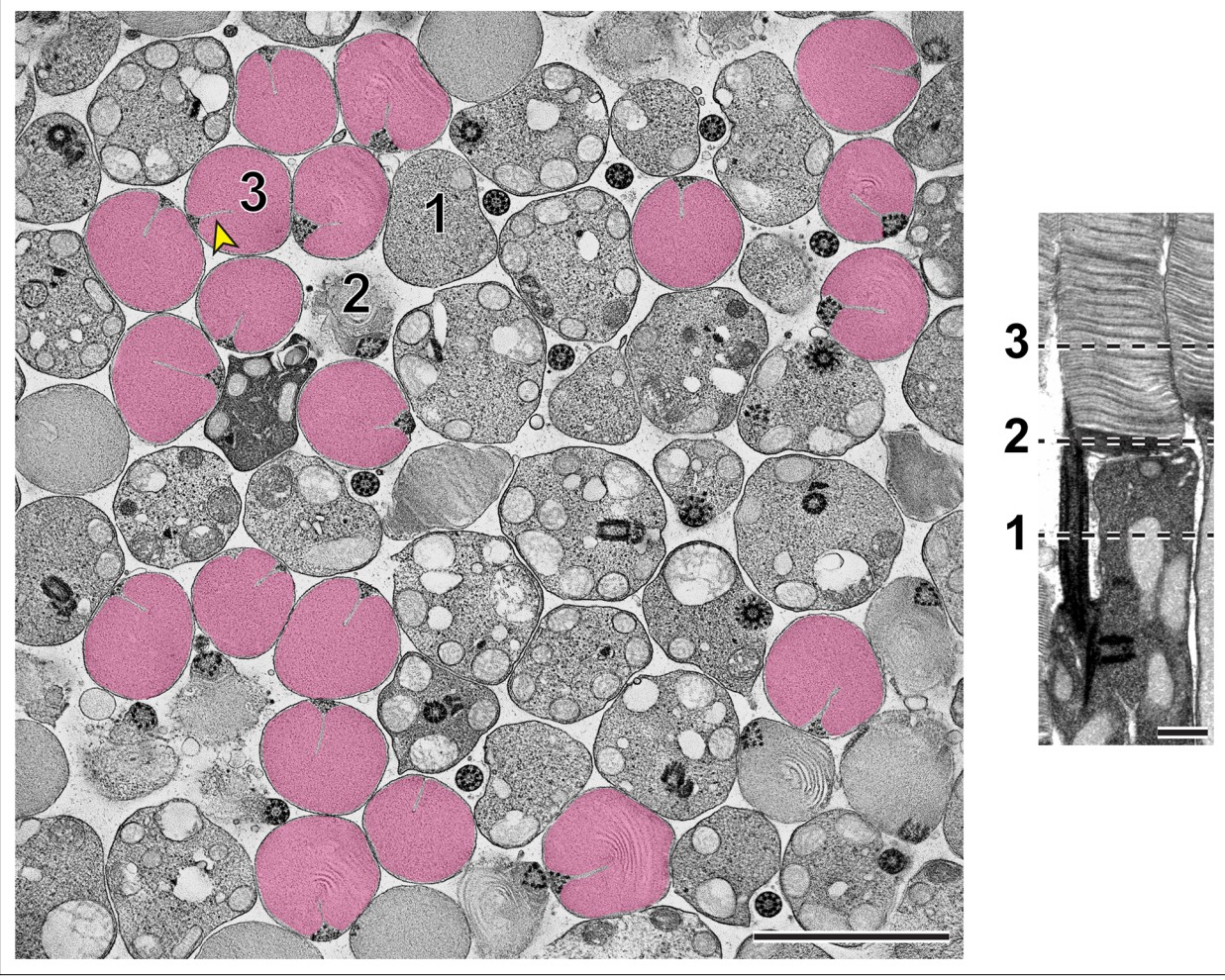

**Figure 3.** Representative *z*-section from a STEM tomogram of a 750-nm-thick tangential section of a WT mouse retina. The retina was contrasted with tannic acid/uranyl acetate. Full tomogram is shown in *Figure 3—video 1*. Because individual rods in the mouse retina are not perfectly aligned, adjacent cells are sectioned at different compartments, including the inner segment (1), outer segment base (2) and mature outer segment (3). Shown on the right is an example of a longitudinally sectioned rod, in which these three locations are depicted by dashed lines. Surfaces of fully enclosed discs are pseudo-colored in magenta to highlight the structure of incisures. Yellow arrowhead points to an incisure in a fully enclosed disc. Tomogram pixel size is 3 nm; scale bars: 2.5 µm (*left*) or 0.5 µm (*right*).

The online version of this article includes the following video for figure 3:

**Figure 3—video 1.** Tomogram associated with *Figure 3*.

https://elifesciences.org/articles/89160/figures#fig3video1

in *Figure 2* reveal that incisures are never found in the newly forming, open discs at the base of the outer segment, but become evident as soon as discs become enclosed. These observations suggest that incisure formation is a process that only occurs after complete disc enclosure.

To fully appreciate the architecture of disc incisures, we conducted electron tomography, which provides a three-dimensional reconstruction of a sample with a resolution of ~1–3 nm in each dimension. Since we were interested in examining the status of disc enclosure and incisure formation in multiple discs at the outer segment base, we utilized a variation of this technique called scanning transmission electron microscopy (STEM) tomography, which allows for the imaging of thicker sections than other electron tomography approaches.

Using STEM tomography, we imaged ~750 nm plastic sections of mouse retinas cut tangentially at the region containing the interface between photoreceptor inner and outer segments. A representative example of a 3D reconstruction of such a section imaged at low magnification is shown in *Figure 3—video 1* and an individual 3 nm *z*-section from this tomogram is shown in *Figure 3*. Because the bases of outer segments from individual photoreceptors are not precisely aligned in the retina, individual cells are sectioned across either inner or outer segments or their junctions. Notably, discernable incisures were observed in outer segments sectioned in the plane of mature, enclosed discs. To better appreciate the appearance of incisures, we pseudo-colored the surfaces of enclosed discs in *Figure 3*. To assess the stage of disc formation at which incisures are formed, we performed higher magnification tomography of individual rods whose outer segment base was in the section. Two major observations from these experiments can be appreciated from the examples shown in *Figure 4* and *Figure 4—videos 1–4*, as follows:

*First* and consistent with TEM data, no incisures were observed in the newly forming discs. *Second*, the newly forming discs are surprisingly lamellipodia-like in shape. While traditionally illustrated as round evaginations of the ciliary plasma membrane (such as in *Steinberg et al., 1980* or *Spencer et al., 2020*), the newly forming discs are actually irregular in shape, oftentimes with multiple protrusions. This observation, best appreciated in the individual *z*-sections shown in the left panels of *Figure 4*, is consistent with the notion that new discs are formed in a process akin to lamellipodia formation mediated by polymerization of a branched actin network (*Spencer et al., 2019*).

Another interesting observation from these tomograms is that incisures always originate from a spot adjacent to the ciliary axoneme. The axoneme itself, being perfectly cylindrical when it emanates from the basal body (*Video 1*), adopts a triangular shape at the spot where incisures are first formed (*Figure 4* and *Figure 4—videos 1–4*). Comparable observations have been previously reported for a variety of species (e.g. *Cohen, 1960*; *Young, 1971*; *Steinberg and Wood, 1975*; *Wen et al., 1982*; *Roof et al., 1991*; *Eckmiller, 2000*). This suggests that microtubules of the ciliary axoneme establish the initial orientation of incisures, although it is unclear whether the transition in axonemal shape may be a part of the mechanism or a consequence of incisure formation.

In another experiment, we performed high-magnification STEM tomography on a part of an outer segment containing mature discs with fully formed incisures (*Figure 5A* and *Figure 5—video 1*). We observed several structural elements (*Figure 5B*), including some that appeared to emanate from a microtubule of the ciliary axoneme and connect to the disc rim at the origin point of an incisure. The regulation of the spacing between discs has been proposed to involve microtubules (*Gilliam et al., 2012*), suggesting that the physical connections that we observed between microtubules and the disc rim could be involved in this process. We also observed structures that appear to connect the two apposing sides of an incisure, which are comparable to structures previously observed in toads (*Roof and Heuser, 1982*). Lastly, we frequently observed electron-dense structures at the incisure ends. Highlighting the power of STEM tomography, we were able to resolve incisures across a span of over 20 discs (*Figure 5C* and *Figure 5—video 1*). Despite always originating from the ciliary axoneme, incisures in adjacent discs were not well-aligned along their entire lengths.

## Incisure size and shape are determined by the relative outer segment contents of peripherin-2 and rhodopsin

In the second part of this study, we explored the role of peripherin-2 in incisure formation. As described above, peripherin-2 plays an essential role in the formation of disc rims and transgenic overexpression of some peripherin-2 mutants in frogs disrupts the structure of incisures. The role of peripherin-2 in incisure formation should be considered in the context of its relative abundance with rhodopsin.

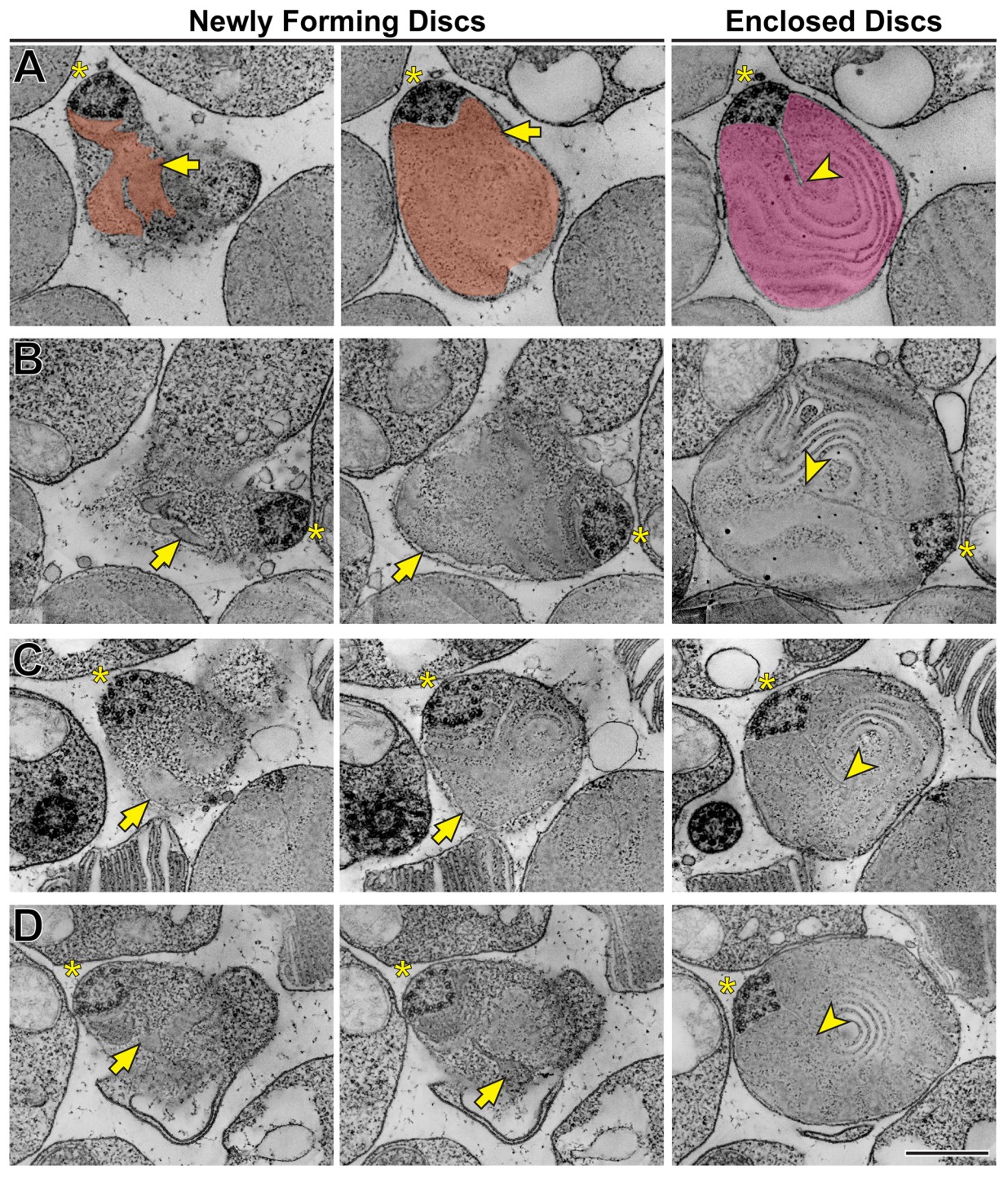

**Figure 4.** Disc incisures are observed in mature but not newly forming discs. (**A**) Representative *z*-sections at the depths of 0 (*left*),+168 (*middle*) and +244 nm (*right*) from a reconstructed electron tomogram of a 750 nm-thick WT mouse retinal section. Full tomogram is shown in *Figure 4—video 1*. To guide the reader, the surfaces of newly forming discs are pseudo-colored in orange and a mature disc in magenta. Pseudo-coloring was used only in this example as it masks some fine morphological features of the images. (**B**) Representative *z*-sections at the depths of 0 (*left*),+107 (*middle*) and +343 nm (*right*) from the reconstructed electron tomogram shown in *Figure 4—video 2*. (**C**) Representative *z*-sections at the depths of 0 (*left*),+65 (*middle*) and +186 nm (*right*) from the reconstructed electron tomogram shown in *Figure 4—video 3*. (**D**) Representative *z*-sections at the depths of 0 (*left*),+19 (*middle*) and +208 nm from the reconstructed electron tomogram shown in *Figure 4—video 4*. Yellow asterisks indicate the ciliary axoneme; yellow arrows point to newly forming, unenclosed discs; yellow arrowheads point to incisures in fully enclosed discs. Tomogram pixel size is 4.2 nm (**A**) or 2.1 nm (**B–D**); scale bar: 0.5 µm.

*Figure 4 continued on next page*

*Figure 4 continued*

The online version of this article includes the following video(s) for figure 4:

**Figure 4—video 1.** Tomogram associated with *Figure 4A*.

https://elifesciences.org/articles/89160/figures#fig4video1

**Figure 4—video 2.** Tomogram associated with *Figure 4B*.

https://elifesciences.org/articles/89160/figures#fig4video2

**Figure 4—video 3.** Tomogram associated with *Figure 4C*.

https://elifesciences.org/articles/89160/figures#fig4video3

**Figure 4—video 4.** Tomogram associated with *Figure 4D*.

https://elifesciences.org/articles/89160/figures#fig4video4

Whereas rhodopsin serves as an essential structural element of the disc lamella, peripherin-2 (along with its homologous partner ROM1) form the disc rim. Together, rhodopsin and peripherin-2/ROM1 comprise ~99% of the total transmembrane protein material in normal discs (*Skiba et al., 2023*). Therefore, we sought to investigate whether the relationship between the surface area of a disc and the length of its rim, including both circumference and incisure, is determined by the molar ratio between rhodopsin and peripherin-2/ROM1.

The knockout of peripherin-2 (which occurs in homozygous *rds* mice *Connell et al., 1991*; *Travis et al., 1991*) is not a useful model to study incisure formation because it completely abolishes outer segment morphogenesis (*Cohen, 1983*; *Jansen and Sanyal, 1984*). However, outer segments are still formed in heterozygous *rds* (*rds/+*) mice despite a reduction in the level of peripherin-2 (*Hawkins et al., 1985*; *Cheng et al., 1997*). Whereas some of these outer segments are dysmorphic due to abnormal disc outgrowths (*Figure 6—figure supplement 1*), others retain their cylindrical shape, at least in young mice. We analyzed the ultrastructure of these relatively normal *rds/+* outer segments in tangential sections using TEM and found a near complete ablation of incisure formation in their discs (*Figure 6A*). In fact, short incisures were present in only two out of 75 analyzed *rds/+* discs, whereas in WT discs, incisures spanned ~50% of the disc diameter in all analyzed outer segments (*Figure 6B*). These observations suggest that a decrease in relative peripherin-2 content in discs suppresses incisure formation.

We next analyzed hemizygous rhodopsin ($Rho^{+/-}$) mice, in which the molar fraction of peripherin-2 in discs is increased due to a reduction in rhodopsin expression. $Rho^{+/-}$ outer segments were previously shown to lack major morphological abnormalities apart from a reduced disc diameter ( +/-), increased incisure length *Makino et al., 2012*; *Price et al., 2012* and somewhat reduced outer segment length (*Liang et al., 2004*; *Price et al., 2012*; see also *Figure 6—figure supplement 1*). Interestingly, we found that disc incisures in these mice were not just longer but also had significant variability in shape. Some of them were mostly straight, as in WT mice, but extended the entire disc diameter (*Figure 7A*). Others were bifurcated or twisted along their length (*Figure 7B*). A subset of outer segments contained tubular structures aligned along the incisure (*Figure 7C*), which are strikingly similar to structures observed in rhodopsin knockout ($Rho^{-/-}$) mice (*Figure 7D*) thought to be formed by peripherin-2 in the absence of normal disc formation (*Chakraborty et al., 2014*). They are also similar to the tubular structures forming in cultured cells expressing recombinant peripherin-2 (*Milstein et al., 2017*; *Salinas et al., 2017*; *Milstein et al., 2020*). Therefore, the tubules observed in some $Rho^{+/-}$ outer segments are likely to be formed by the excess of peripherin-2 not incorporated into the incisure. These observations

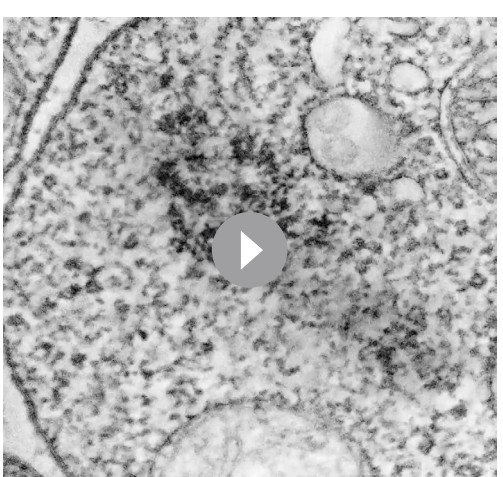

**Video 1.** Reconstructed tomogram of a basal body nucleating the ciliary axoneme. Shown is a 420 nm fragment of a 750-nm-thick retinal section. Tomogram pixel size is 0.7 nm. Field of view: 0.80 μm x 0.80 μm.

https://elifesciences.org/articles/89160/figures#video1

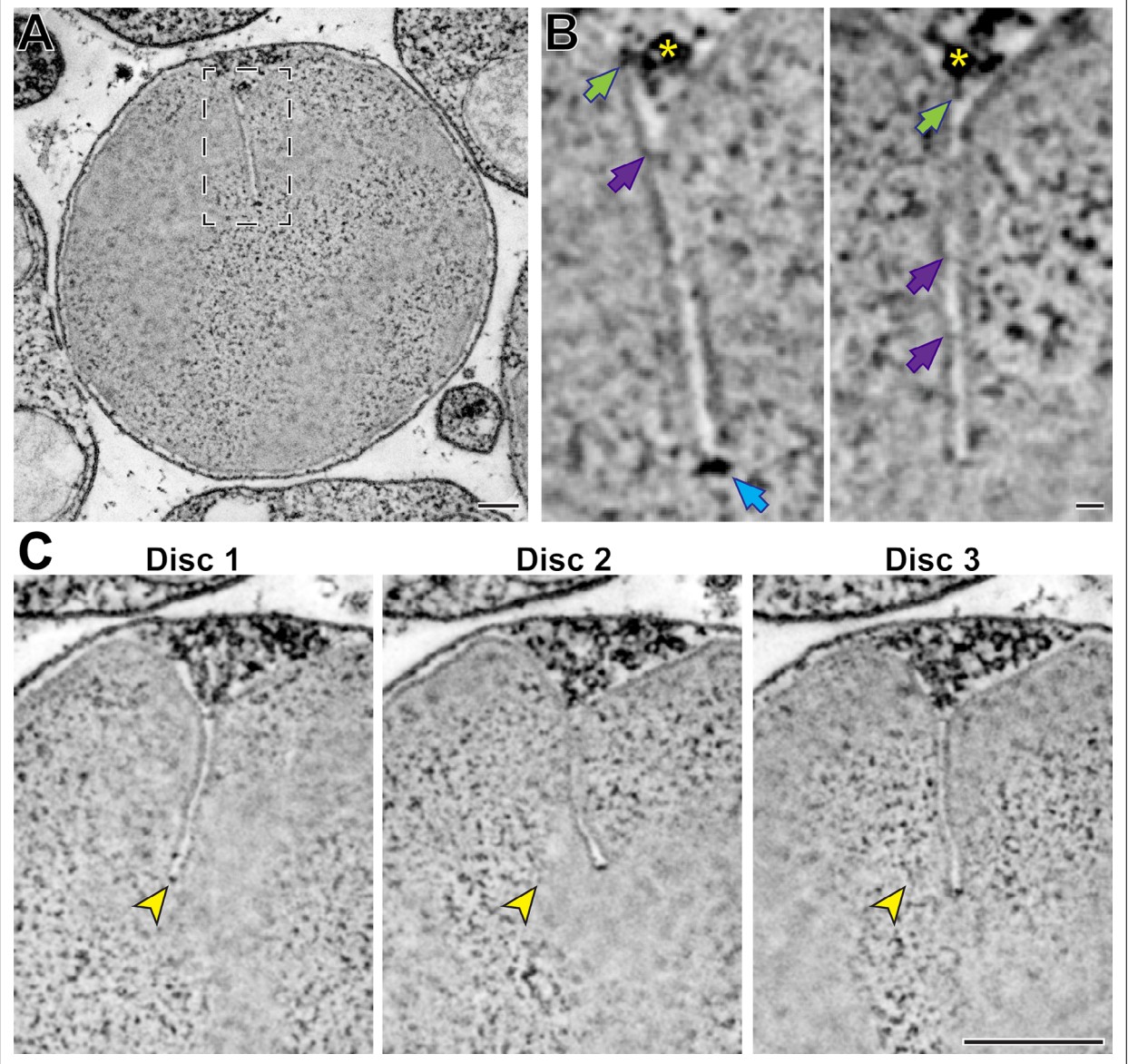

**Figure 5.** Ultrastructure of mature disc incisures. (**A**) Representative *z*-section from a reconstructed electron tomogram of a 750-nm-thick WT mouse retinal section. Full tomogram of the boxed area is shown in *Figure 5—video 1*. (**B**) Representative maximum intensity projections of 3 (*left*) or 4 (*right*) *z*-sections highlighting various structures observed in a mature incisure. Green arrows point to structures spanning between microtubules and the disc rim; purple arrows point to connectors between apposing sides of the incisure; blue arrow points to the electron-dense structure at the incisure end. Yellow asterisk indicates the microtubule adjacent to the incisure. (**C**) Representative *z*-sections at the depths of 0 (Disc 1),+31.5 (Disc 2) and +63 nm (Disc 3) illustrating an imperfect alignment of incisures in three adjacent discs. Yellow arrowheads point to the incisure end in Disc 1 and the same *x,y*-coordinates in subsequent discs. Tomogram pixel size is 2.1 nm; scale bars: 0.25 μm (**A, C**) or 25 nm (**B**).

The online version of this article includes the following video for figure 5:

**Figure 5—video 1.** Tomogram associated with *Figure 5*.

https://elifesciences.org/articles/89160/figures#fig5video1

suggest that an increase in relative peripherin-2 content in discs causes an increase in incisure size and complexity.

To further explore the idea that the sizes of discs and incisures are defined by the molar ratio between rhodopsin and peripherin-2, we measured this ratio in outer segments obtained from mice of all three genotypes. We also analyzed the levels of ROM1 because it is another tetraspanin contributing to disc rim formation by oligomerizing with peripherin-2. We used a quantitative mass

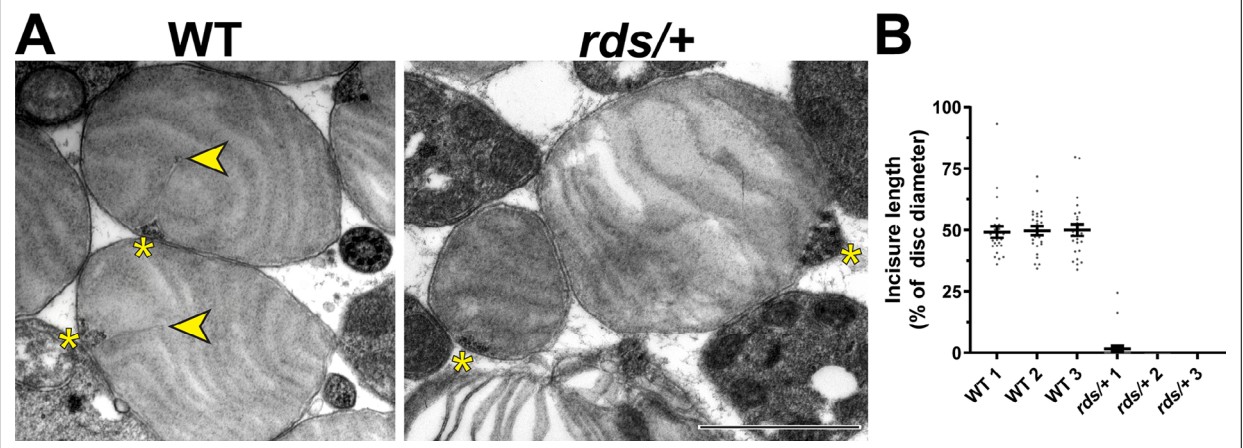

**Figure 6.** Reduction in peripherin-2 level prevents incisure formation in mouse rods. (**A**) Representative TEM images of tangentially sectioned WT outer segments and *rds/+* outer segments preserving their cylindrical shape. Yellow asterisks indicate the ciliary axoneme; yellow arrowheads point to incisures in WT discs. Scale bar: 1 µm. (**B**) Quantification of incisure length as a percent of the total disc diameter. Each data point represents a single outer segment. For each genotype, three mice were analyzed (labeled as 1, 2, and 3), with 25 outer segments analyzed in each mouse. Only 2 out of 75 analyzed *rds/+* rods contained discernible incisures. Error bars represent mean ± s.d.

The online version of this article includes the following figure supplement(s) for figure 6:

**Figure supplement 1.** Reduction in the level of peripherin-2 but not rhodopsin causes gross abnormalities in outer segment structure.

spectrometry approach that we have recently applied to determine the molar ratio amongst multiple outer segment proteins, including these three proteins (*Skiba et al., 2023*). Our measurements are summarized in *Table 1* (see for raw data).

In WT outer segments, the molar ratios of rhodopsin to peripherin-2 and ROM1 were ~18:1 and ~42:1, respectively, as in *Skiba et al., 2023*. In *rds/+* outer segments, the rhodopsin to peripherin-2 ratio was increased by ~1.7 fold (~31:1), whereas in *Rho*[+/-] outer segments it was decreased by +/-2 fold (~9:1). Both changes are consistent with these mouse lines expressing single copies of the corresponding gene (see also *Lem et al., 1999*; *Chakraborty et al., 2014*). Considering ROM1, *rds/+* outer segments contained relatively more ROM1 than WT outer segments (rhodopsin to ROM1 ratios of ~30:1 and ~42:1, respectively), suggesting that more ROM1 is incorporated into disc rims when there is a deficiency in peripherin-2. Accordingly, the molar ratio between peripherin-2 and ROM1 shifted from ~2.3:1 in WT rods to ~1:1 in *rds/+* rods. In contrast, the molar ratio between peripherin-2 and ROM1 was essentially unaffected in *Rho*[+/-] +/- (~2.6:1 vs. ~2.3:1 in WT rods), indicating that rhodopsin deficiency does not disrupt the balance between these two tetraspanins.

Overall, the molar ratio between rhodopsin and total tetraspanin protein (peripherin-2 and ROM1 combined) changed from ~13:1 in WT rods to ~15:1 in *rds/+* rods and ~6:1 in *Rho*[+/-] +/-. These measurements are consistent with our qualitative conclusions from the ultrastructural analysis of these mice and suggest that these conclusions apply to both ratios of rhodopsin to peripherin-2 and the total tetraspanin content.

In the case of *Rho*[+/-] +/- whose entire outer segment population is devoid of major morphological defects, we sought to determine whether the measured increase in relative tetraspanin abundance could explain the observed increase in the size and complexity of incisures. We calculated the theoretical incisure length in *Rho*[+/-] +/- based on the assumptions that the disc surface area is proportional to the number of rhodopsin molecules, whereas the entire length of the disc rim (including both the circumference and incisure) is proportional to the number of tetraspanin molecules. Unfortunately, *rds/+* mice could not be analyzed this way because the majority of their outer segments did not produce proper discs (*Figure 6—figure supplement 1*).

We measured disc diameters in WT and *Rho*[+/-] +/- to be ~1.5 and ~1.1 µm, respectively (*Figure 8*), which is consistent with previous reports (*Liang et al., 2004*; *Makino et al., 2012*; *Price et al., 2012*; *Lewis et al., 2020*). These values correspond to disc surface areas of 3.53 and 1.80 µm², respectively (see *Table 2* for all values calculated in this analysis). We next estimated the number of rhodopsin and tetraspanin molecules in a single disc of each genotype. Given that an average WT mouse rod

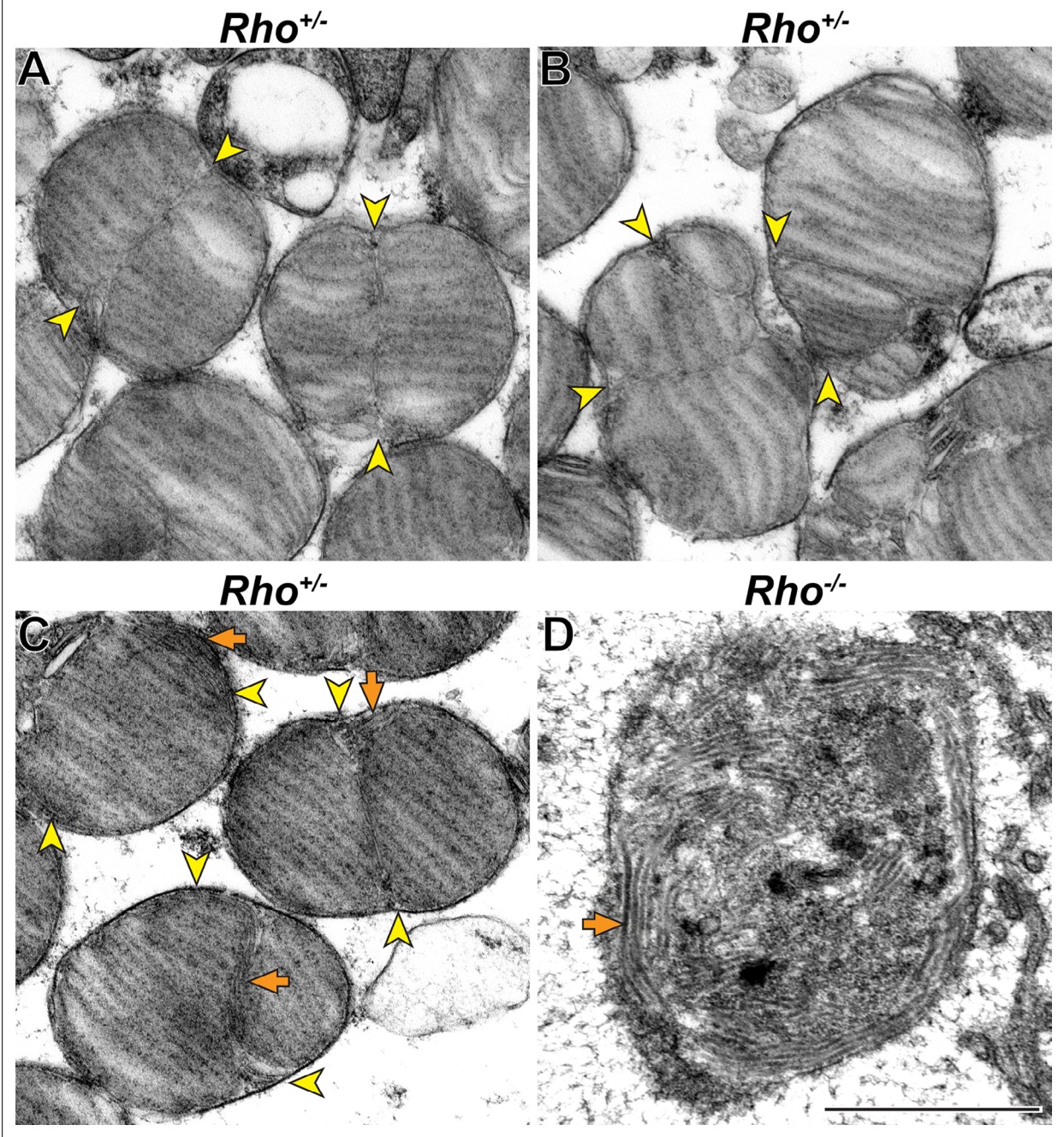

**Figure 7.** Increase in relative peripherin-2 level produces long incisures of varying shape. (**A–C**) Representative TEM images of tangentially sectioned *Rho*$^{+/-}$ mouse retinas. Incisures can be relatively straight and extend nearly the entire disc diameter (**A**) or be bifurcated or twisted (**B**). In some cells, incisures are associated with tubular structures (**C**). (**D**) Representative TEM image of a tangentially sectioned *Rho*$^{-/-}$ retina. While lacking discs, the outer segment cilium contains a large number of tubular structures. Yellow arrowheads point to incisure ends; orange arrows point to tubular structures. Scale bar: +/- μm.

contains ~60 million rhodopsin molecules (*Lyubarsky et al., 2004*; *Nickell et al., 2007*) and ~800 discs (*Liang et al., 2004*), there are ~75,000 rhodopsin molecules in each disc. Considering that rhodopsin packing density in disc membranes is unaffected by its expression level (*Liang et al., 2004*), we calculated that the number of rhodopsin molecules in *Rho*$^{+/-}$ +/- is~38,200, based on its surface area. Using our measured rhodopsin to tetraspanin ratios, we determined that the total number of tetraspanin molecules per disc in WT and *Rho*$^{+/-}$ +/- is similar and equal to ~5910 and~6360, respectively.

**Table 1.** Quantification of molar ratios between rhodopsin, peripherin-2 and ROM1 in mouse outer segments.

| Protein molar ratio* | WT | rds/+ | Rho⁺/⁻ |
|---|---|---|---|
| Rhodopsin: peripherin-2 | 18.2±0.6 | 30.9±6.0 | 8.5±1.5 |
| Rhodopsin: ROM1 | 42.2±0.6 | 29.9±2.0 | 21.6±3.7 |
| Peripherin-2: ROM1 | 2.3±0.1 | 1.0±0.2 | 2.6±0.6 |
| Rhodopsin: (Peripherin-2 +ROM1) | 12.7±0.4 | 15.1±1.8 | 6.0±0.8 |

*Values are shown as mean ± s.d. Three outer segment preparations from mice of each genotype were analyzed.

The online version of this article includes the following source data for table 1:

**Source data 1.** Quantification of molar ratios between rhodopsin, peripherin-2 and ROM1 in mouse outer segments – raw data.

Next, we measured the total disc rim length in WT discs, including the incisure, and found it to be 6.21 µm. Assuming that all tetraspanin molecules in fully enclosed discs are located at the rims, we estimated that there are ~950 tetraspanin molecules per 1 µm of disc rim in WT rods. Assuming that this tetraspanin density is not changed in disc rims of $Rho^{+/-}$ +/-, we calculated that the total rim length in $Rho^{+/-}$ +/- is predicted to be ~6.69 µm. Given that the circumference of $Rho^{+/-}$ +/- is~3.36 µm, the predicted incisure length is ~1.66 µm, or ~155% of the disc diameter. The fact that this value exceeds 100% is consistent with the observed abnormalities in incisure length and shape, including bifurcated or twisted incisures and adjacent tubular structures. Of note, the outer segment length does not affect this analysis because the molar ratio between rhodopsin and tetraspanins in each disc is invariant across the entire disc stack.

## Peripherin-2 knockout frogs form outer segments with greatly reduced incisures

In the last set of experiments, we explored the role of peripherin-2 in controlling incisure formation in frog photoreceptors, as their discs contain more than a single incisure (*Figure 1*). We generated a peripherin-2 knockout (*prph2⁻/⁻*) *Xenopus tropicalis* frog using CRISPR-Cas9 to create mutations within exon 1 of *prph2* (see Materials and methods for additional details). Ultrastructural analysis of their retinas revealed a relatively normal outer segment morphology, except for occasional overgrowth of disc membranes sometimes shaped as whorls (*Figure 9*).

The formation of outer segments in these frogs was surprising considering the complete lack of outer segments in peripherin-2 knockout mice. This cross-species difference could be explained by *Xenopus* expressing multiple peripherin-2 homologs. It was reported that *Xenopus laevis* expresses three peripherin-2 homologs termed xrds38, xrds36, and xrds35 (*Kedzierski et al., 1996*). The current annotations of the *Xenopus tropicalis* and *Xenopus laevis* genomes suggest that xrds38 (knocked out in our current study) is the frog peripherin-2 homolog (XenBase: XB-GENEPAGE-985593) while xrds36 and xrds35 are isoforms of ROM1 (XenBase: XB-GENEPAGE-962405). In addition, these genomes contain another peripherin-2-like gene, termed *prph2l*

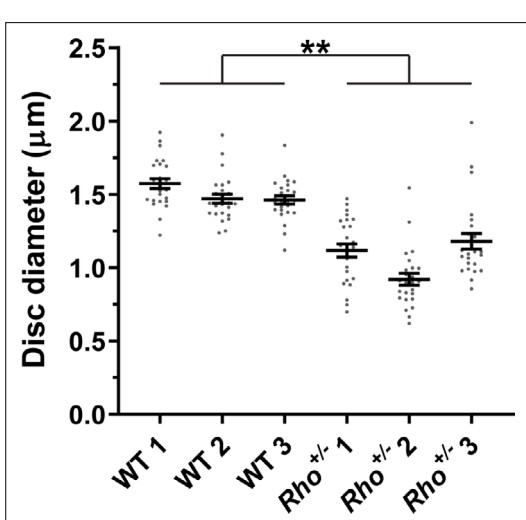

**Figure 8.** Quantification of disc diameters in WT and $Rho^{+/-}$ mouse rods. Each data point represents a single outer segment. For each genotype, three mice were analyzed (labeled as 1, 2, and 3), with 25 outer segments analyzed in each mouse. Error bars represent mean ± s.d. Unpaired t-test was performed using the average disc diameter in each mouse to determine that the difference in diameters of WT and $Rho^{+/-}$ +/- was statistically significant (p=0.0075).

**Table 2.** Summary of quantitative parameters determined for mouse discs.

| Calculated parameter | WT | Rho[+/-] |
|---|---|---|
| Disc surface area (both lamellae) | 3.53 µm² | 1.80 µm² |
| Rhodopsin molecules per disc | 75,000 | 38,200 |
| Peripherin-2+ROM1 molecules per disc | 5,910 | 6,360 |
| Total rim length | 6.21 µm | N/A* |
| Measured incisure length as % of disc diameter | 50%† | N/A* |
| Theoretical incisure length as % of disc diameter | - | 155% |

*Incisure and total rim lengths in *Rho +/-* discs cannot be readily measured due to incisure complexity.
†The value is taken from the measurements shown in **Figure 6**.

(XenBase: XB-GENEPAGE-5759091). Therefore, it is plausible that peripherin-2-like protein and/or ROM1 isoforms are sufficient or even play a primary role in supporting outer segment morphogenesis in frog photoreceptors.

As previously shown, discs of WT frogs contained numerous incisures (*Figure 9*). Like in mice, one of these incisures was aligned with the ciliary axoneme (*Figure 9—figure supplement 1*). In contrast, rods of *prph2⁻/⁻* frogs displayed a nearly complete lack of incisures (*Figure 9*), apart from rare examples of discs containing a single incisure (*Figure 9F*). This phenotype resembles that of *rds/+* mice and shows that peripherin-2 is an important contributor to the formation of the entire disc rim structure in frogs that cannot be fully replaced by other homologous tetraspanins expressed in these cells.

## Discussion

In this study, we explored the mechanisms underlying the formation of disc incisures in rod photoreceptors and report two central findings. *First*, we found that incisures are formed only after the completion of disc enclosure. Notably, this is contrary to the conclusion in the classical paper by Steinberg and colleagues which first described the currently accepted mechanism of disc morphogenesis (*Steinberg et al., 1980*). The authors concluded that "incisure formation occurs … before rim formation is complete around the entire perimeter of the disc … Closure of the disc, therefore, is not a prerequisite for incisure formation". However, the structures in nascent discs that were interpreted as incisures in Figure 11 of *Steinberg et al., 1980* were not aligned longitudinally, which is a hallmark property of incisures. Such a lack of alignment suggests an alternative explanation for the small discontinuities of disc membranes that they interpreted as incisures. These discontinuities could be explained by a thin plastic section being cut through an uneven edge of an expanding disc, which created an appearance of a gap between the most expanded portions of the disc lamella. How uneven nascent disc expansions can be prior to assuming their final round shape is well-illustrated in the 3D tomograms presented in our study. A related argument was made in another study employing 3D tomography (*Volland et al., 2015*), which illustrated that, when analyzed in thin sections, enclosing discs may artificially appear as intracellular vesicles.

It is generally accepted that the formation of incisures is limited to rods (*Goldberg et al., 2016*). This could be appreciated in *Figure 9A* showing two cones lacking incisures alongside rods having multiple incisures in a WT frog retina. Because mature cone discs do not enclose or enclose partially in mammals, the lack of cone incisures is consistent with our conclusion that incisure formation takes place after disc enclosure. However, we must note that there have been a few reports of mammalian cones with a single incisure (*Steinberg and Wood, 1975*; *Anderson and Fisher, 1979*; *Carter-Dawson and LaVail, 1979*), which warrants further investigation into the status of disc enclosure and the presence of incisures in mammalian cones.

Our *second* central finding is that incisure size and complexity are dependent on the relative outer segment contents of rhodopsin and peripherin-2. An excess of peripherin-2 leads to longer, more complex incisures, whereas a deficiency precludes incisure formation along with an occasional inability of discs to enclose. Less clear is the exact contribution of ROM1 to these processes. In the case of *rds/+* mice, characterized by a rather severe defect in outer segment morphogenesis, the ratio of

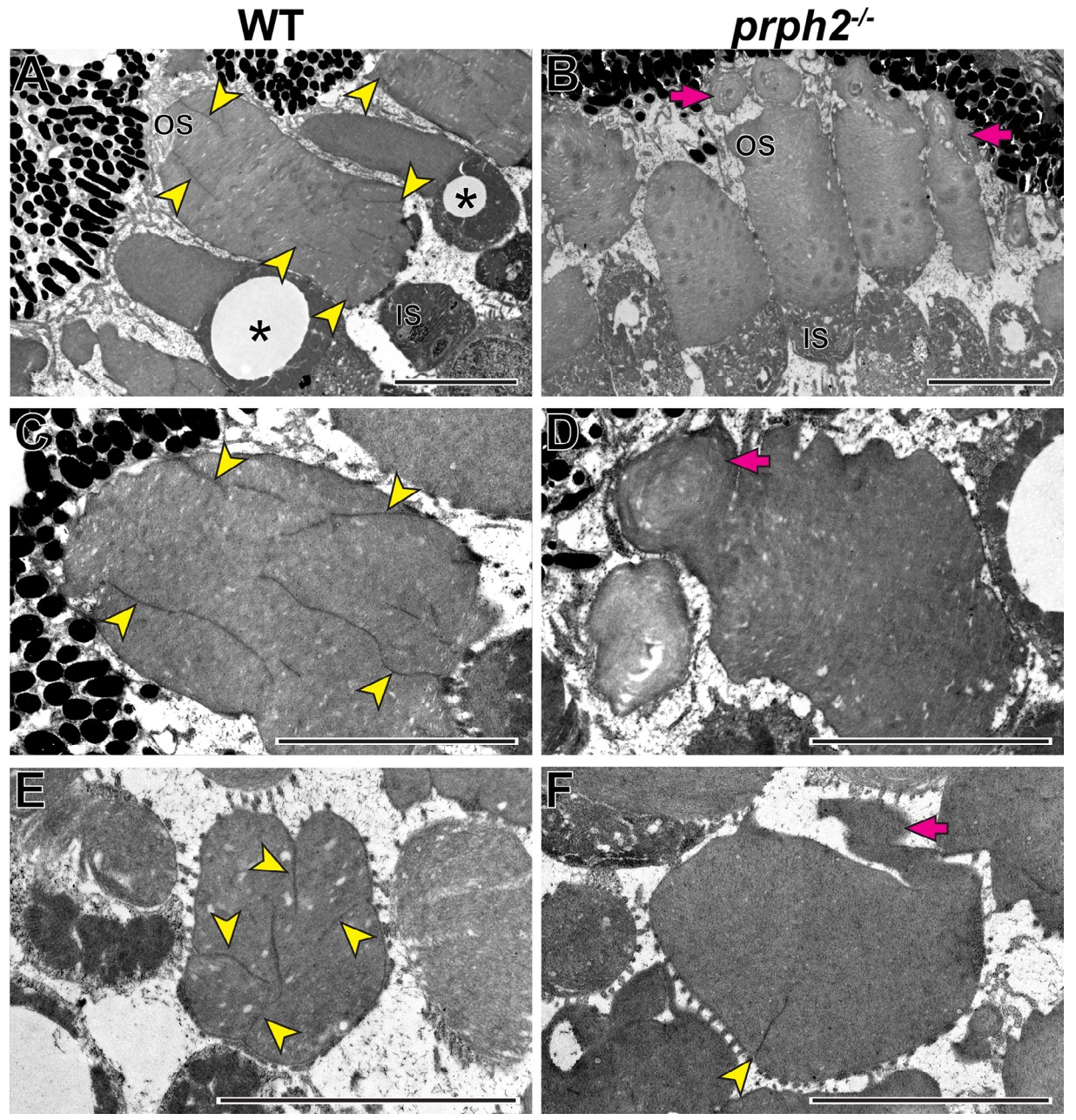

**Figure 9.** Loss of peripherin-2 in frogs (*Xenopus tropicalis*) prevents incisure formation. (**A–D**) Representative TEM images of longitudinally sectioned WT and *prph2⁻/⁻* frog retinas. (**E,F**) Representative TEM images of tangentially sectioned WT and *prph2⁻/⁻* frog retinas. Magenta arrows point to defects in outer segment morphology; yellow arrowheads point to incisures; asterisks indicate cones, as evident by the presence of an oil droplet in their inner segments. OS: outer segment; IS: inner segment. Scale bars: 5 µm.

The online version of this article includes the following figure supplement(s) for figure 9:

**Figure supplement 1.** One incisure in frog rod discs is aligned with the ciliary axoneme.

rhodopsin to total tetraspanin protein in disc membranes decreases only modestly compared to WT mice (~15:1 vs. ~13:1) due to an increase in ROM1 content. It is intriguing to speculate that the gross morphological defects of *rds/+* discs may not be caused by this modest reduction in total tetraspanin content, but rather by the nearly 2-fold reduction in the relative peripherin-2 content. This would argue that ROM1 is unable to efficiently compensate for a deficiency in peripherin-2. Consistent with this line of thought, *Rom1⁻/⁻* outer segments have relatively minor morphological defects (*Clarke et al.,*

*2000*) despite ROM1 accounting for ~30% of the total tetraspanin content of the outer segment. The specific role of ROM1 in supporting disc structure remains to be determined in future experiments.

Taken together, these findings led us to propose that rod photoreceptors have evolved to express a slight excess of peripherin-2 over the amount needed to fully enclose their discs. Ensuring full enclosure is important because defects in this process lead to uncontrolled disc membrane outgrowth and ultimate photoreceptor cell death. Such a pathology is observed in photoreceptors with a reduced molar ratio of peripherin-2 to rhodopsin, such as in *rds/+* mice (*Hawkins et al., 1985*; *Sanyal et al., 1986*; *Sanyal and Hawkins, 1986*) and in mice overexpressing rhodopsin (*Price et al., 2012*). The same pathology is caused by mutations in peripherin-2 that affect its oligomerization (*Stuck et al., 2014*; *Zulliger et al., 2018*; *Conley et al., 2019*; *Milstein et al., 2020*; *Lewis et al., 2021*). Unlike peripherin-2 deficiency, an excess of peripherin-2 does not lead to severe morphological outer segment defects, such as in $Rho^{+/-}$ +/- or mice overexpressing peripherin-2 (*Nour et al., 2004*). These differences could be appreciated in the side-by-side comparison of *rds/+* and $Rho^{+/-}$ outer segments in +/-. Therefore, the fact that rods express a slight excess of peripherin-2 over the amount required for disc enclosure may be viewed as an evolutionary adaptation to prevent severe pathology arising from incomplete disc enclosure. Following the completion of disc enclosure, this excess of peripherin-2 is deposited in the form of an incisure, which may serve to protect the flat lamellar disc membranes from undesirable deformations.

A second, not mutually exclusive function of incisures that has been long-discussed in the literature is to promote longitudinal diffusion of signaling molecules, such as cGMP and $Ca^{2+}$, in the outer segment (*Ichikawa, 1996*; *Holcman and Korenbrot, 2004*; *Caruso et al., 2006*; *Bisegna et al., 2008*; *Gross et al., 2012*; *Makino et al., 2012*; *Caruso et al., 2020*). This may increase the sensitivity and uniformity of light responses, particularly those evoked by a small number of photons. Theoretical analysis shows that this mechanism is particularly relevant in wider rods containing multiple incisures, but may be less significant in thin rods containing a single incisure, such as in the mouse (*Caruso et al., 2006*; *Caruso et al., 2020*). These conclusions remain to be tested in physiologically intact photoreceptors.

Another unanswered question is how the number of incisures in a disc is determined. While mouse rods have a single incisure, amphibian rods may have over 20. It is unlikely that incisure number is regulated merely by the relative tetraspanin content, as we typically observed elongated incisures rather than multiple incisures in $Rho^{+/-}$ +/-. It is likely that incisure number is regulated by the abundance of outer segment microtubular structures. In fact, the presence of non-axonemal microtubules aligned with incisures was documented in frog rods (*Eckmiller, 2000*). The goal of future studies is to elucidate the specific role of microtubules in controlling incisure formation and number.

## Materials and methods

### Key resources table

| Reagent type (species) or resource | Designation | Source or reference | Identifiers | Additional information |
|---|---|---|---|---|
| strain, strain background (*Mus musculus*) | C57BL/6 J | Jackson Labs | Jax#:000664 | |
| genetic reagent (*Mus musculus*) | *Rho* | *Lem et al., 1999* | MGI:2680822 | |
| genetic reagent (*Mus musculus*) | *rds* | *van Nie et al., 1978* | MGI:1856523 | |
| strain, strain background (*Xenopus tropicalis*) | Nigerian | National *Xenopus* Resource | RRID:NXR_1018 | |
| genetic reagent (*Xenopus tropicalis*) | *prph2* | This paper | RRID:NXR_3003 | National *Xenopus* Resource |

### Animal husbandry

Animal maintenance and experiments were approved by the Institutional Animal Care and Use Committees at Duke (Durham, NC; protocol #A184-22-10) and the Marine Biological Laboratory (Woods Hole, MA; protocol #22–29). WT mice (*Mus musculus*) were C57BL/6 J (Jackson Labs stock #000664). *rds* mice are described in *van Nie et al., 1978*. $Rho^{-/-}$ mice are described in *Lem et al., 1999*. Mice were genotyped to ensure that they did not contain either the *rd8* (*Mattapallil et al., 2012*) or *rd1* (*Pittler et al., 1993*) mutations commonly found in inbred mouse strains. *prph2* mutant frogs (*Xenopus tropicalis*) are described below. Mice were used at 1 month of age. Frogs were used

at either 7- or 14 days post fertilization. All experiments were performed with animals of randomized sex and, for each experiment, at least three biological replicates were analyzed.

## Generation of *prph2* knockout *Xenopus tropicalis*

The *prph2* knockout line (RRID:NXR_3003) was generated using CRISPR-Cas9. CRISPRScan *Moreno-Mateos et al., 2015* was used to design two sgRNAs within the first exon of *prph2* (sgRNA1: GGGGTC TGCTTCTTGGCCAG; sgRNA2: GGGATACTGACACCCCCGGC) with 5' dinucleotides converted to GG for increased mutagenic activity (*Gagnon et al., 2014*). sgRNAs were synthesized using the SP6 MEGAscript SP6 Transcription Kit (Invitrogen, Waltham, MA). F0 injections were performed by injecting one-cell stage embryos from the *X. tropicalis* Nigerian strain (RRID: NXR_1018). Each embryo was injected with 500 pg each of sgRNA1/sgRNA3 and 1000 pg Cas9. Founders were raised to sexual maturity and screened for germline transmission of mutations. A –20 bp mutation, which results in a frameshift mutation at amino acid 183 and a stop codon 7 amino acids downstream, was selected for generating the *prph2* line. Intercrosses were done on –20 bp heterozygotes to generate *prph2* mutants.

## Tissue fixation

For mice, tissue fixation was performed as described previously (*Ding et al., 2015*). In the morning after lights were turned on, anesthetized mice were transcardially perfused with 2% paraformaldehyde, 2% glutaraldehyde and 0.05% calcium chloride in 50 mM MOPS (pH 7.4) resulting in exsanguination. Enucleated eyes were fixed for an additional 2 hr in the same fixation solution at room temperature prior to processing. For frog tadpoles, tissues were fixed with 2% paraformaldehyde, 2% glutaraldehyde in 0.1 M sodium cacodylate buffer (pH 7.2) overnight at 4 °C.

## Tissue processing

For processing of longitudinal sections of mouse retinas, eyecups were dissected from fixed eyes, embedded in 2.5% low-melt agarose (Precisionary, Greenville, NC) and cut into 200-µm-thick slices on a Vibratome (VT1200S; Leica, Buffalo Grove, IL). Agarose sections were treated with 1% tannic acid (Electron Microscopy Sciences, Hartfield, PA) and 1% uranyl acetate (Electron Microscopy Sciences), gradually dehydrated with ethanol and infiltrated and embedded in Spurr's resin (Electron Microscopy Sciences). For processing of tangential sections of mouse retinas, dissected retinas were treated with 1% tannic acid (Electron Microscopy Sciences) and 1% uranyl acetate (Electron Microscopy Sciences), gradually dehydrated with ethanol and infiltrated and embedded in Spurr's resin (Electron Microscopy Sciences). For processing of frog samples, tadpoles were treated with 1% tannic acid (Electron Microscopy Sciences) and 1% uranyl acetate (Electron Microscopy Sciences), gradually dehydrated with ethanol and infiltrated and embedded in Spurr's resin (Electron Microscopy Sciences).

## Transmission electron microscopy

A total of 70 nm sections were cut from resin-embedded samples, placed on copper grids and counterstained with 2% uranyl acetate and 3.5% lead citrate (Ted Pella, Redding, CA). Samples were imaged on a JEM-1400 electron microscope (JEOL, Peabody, MA) at 60 kV with a digital camera (BioSprint; AMT, Woburn, MA). Image analysis and processing was performed with ImageJ.

## Electron tomography

Of the central retina, 750 nm sections were cut from resin-embedded samples and placed on 50 nm Luxel film slot grids. The grids were glow-discharged on both sides, and a mixture of 10 nm, 20 nm, and 60 nm gold particles were deposited on the sample surfaces to serve as fiducial markers. Electron tomography was conducted on a Titan Halo (FEI, Hillsboro, OR) operating at 300 kV in STEM mode. A 4-tilt series data acquisition scheme previously described (*Phan et al., 2017*) was followed in which the specimen was tilted from –60° to +60° every 0.25° at 4 evenly distributed azimuthal angle positions. The micrographs were collected with a high-angle annular dark field (HAADF) detector. The final volumes were generated using an iterative reconstruction procedure (*Phan et al., 2017*). Image analysis and processing was performed with 3dmod and ImageJ.

## Quantitative mass spectrometry

A crude preparation of rod outer segments was obtained from dissected mouse retinas that had been vortexed in 8% OptiPrep in mouse Ringer's solution (containing 130 mM NaCl, 3.6 mM KCl, 2.4 mM MgCl$_2$, 1.2 mM CaCl$_2$, and 10 mM HEPES, pH 7.4) that was adjusted to 314 mOsm. The preparation was briefly left on ice to allow the remaining retinal tissue to settle. The supernatant was removed and centrifuged at 20,000 x *g*. Pelleted outer segments were gently washed with mouse Ringer's solution before lysis with 2% SDS in PBS. Protein concentration was measured with the Bio-Rad Protein Assay kit (Bio-Rad, Hercules, CA). Samples containing 20 µg of protein were mixed with 0.5 µg BSA (used as an internal standard in this analysis) and cleaved with 1 µg trypsin/LysC mix (Promega, Madison, WI) using the SP3 beads protocol described in *Hughes et al., 2014*. The combined digest of outer segments and BSA was mixed with the digest of a chimeric protein consisting of concatenated tryptic peptides of outer segment proteins, including rhodopsin, peripherin-2 and ROM1, which is described in *Skiba et al., 2023*. Mass spectrometry, data processing and data analysis were also performed as described in *Skiba et al., 2023*. For each genotype, a total of three biological replicates were analyzed.

## Materials availability statement

The *prph2* knockout frog (RRID:NXR_3003) is available at the National *Xenopus* Resource.

## Acknowledgements

The authors would like to thank Joseph Besharse and Dean Bok for helpful discussions during the course of this study. This work was supported by the NIH grants R01 EY030451 (VYA), P30 EY005722 (VYA), K99 EY033763 (TRL), P40 OD010997 (MEH), R24 OD030008 (MEH) and an Unrestricted Award from Research to Prevent Blindness Inc (Duke University). Electron tomography data acquisition, reconstruction and computer graphic segmentation and display were performed at the National Center for Microscopy and Imaging Research, with support from NIH grant U24 NS120055 (MHE). Deposition and management of acquired raw and derived electron tomography data within the Cell Image Library was further supported by NIH grant R01 GM82949 (MHE). The funders had no role in study design, data collection and interpretation, or the decision to submit the work for publication.

## Additional information

### Funding

| Funder | Grant reference number | Author |
|---|---|---|
| National Institutes of Health | EY030451 | Vadim Y Arshavsky |
| National Institutes of Health | EY005722 | Vadim Y Arshavsky |
| National Institutes of Health | EY033763 | Tylor R Lewis |
| National Institutes of Health | OD010997 | Marko E Horb |
| National Institutes of Health | OD030008 | Marko E Horb |
| Research to Prevent Blindness | Unrestricted Award | Vadim Y Arshavsky |
| National Institutes of Health | NS120055 | Mark H Ellisman |
| National Institutes of Health | GM82949 | Mark H Ellisman |

| Funder | Grant reference number | Author |
|--------|------------------------|--------|

The funders had no role in study design, data collection and interpretation, or the decision to submit the work for publication.

## Author contributions

Tylor R Lewis, Conceptualization, Formal analysis, Funding acquisition, Investigation, Visualization, Methodology, Writing – original draft, Writing – review and editing; Sebastien Phan, Formal analysis, Investigation, Visualization, Methodology, Writing – review and editing; Carson M Castillo, Kelsey Coppenrath, William Thomas, Ying Hao, Investigation, Writing – review and editing; Keun-Young Kim, Investigation, Visualization, Writing – review and editing; Nikolai P Skiba, Formal analysis, Investigation, Writing – review and editing; Marko E Horb, Resources, Supervision, Funding acquisition, Methodology, Writing – review and editing; Mark H Ellisman, Resources, Formal analysis, Supervision, Funding acquisition, Methodology, Project administration, Writing – review and editing; Vadim Y Arshavsky, Conceptualization, Resources, Formal analysis, Supervision, Funding acquisition, Methodology, Project administration, Writing – review and editing

## Author ORCIDs

Tylor R Lewis ⓘ https://orcid.org/0000-0001-6832-7972
Vadim Y Arshavsky ⓘ https://orcid.org/0000-0001-8394-3650

## Ethics

Animal maintenance and experiments were approved by the Institutional Animal Care and Use Committees at Duke (Durham, NC; protocol #A184-22-10) and the Marine Biological Laboratory (Woods Hole, MA; protocol #22-29).

## Decision letter and Author response

Decision letter https://doi.org/10.7554/eLife.89160.sa1
Author response https://doi.org/10.7554/eLife.89160.sa2

# Additional files

## Supplementary files

• MDAR checklist

## Data availability

All data generated or analyzed for this study are included in the manuscript and supporting files.

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
