## [Editor Report]

This study describes the importance of the stoichiometric relationship between rhodopsin and peripherin-2 in mouse rod outer segments. The authors use genetic manipulations to vary the ratio of these two proteins to demonstrate that excess peripherin leads to excess perimeter, which then leads to the infolded structures known as incisures. These data illustrate a fundamental principle that relates factors that control the area of a structure to factors that control the perimeter of a structure.

---

## [Decision Letter]

[Editors' note: this paper was reviewed by Review Commons.]

---

## [Author Response]

General Statements [optional]

We are very pleased that our manuscript was well-received by each of the three reviewers. Reviewer #1 found our study to be “interesting”, “clear” and “convincing”. Reviewer #2 found our study to be “unprecedented” and of “considerable interest to cell biologists in the vision community and likely to the broader cell biology community”. Reviewer #3 believed that we reported “novel and significant findings into an important cell biological problem” and that our study “should be of broad interest to cell biologists and vision scientists”. This reviewer also stated that “it is strongly recommended for publication”.

Point-by-point description of the revisionsReviewer #1:The paper of Lewis et al. presents an interesting study describing new information about the morphology of nascent discs and the role of peripherin in determining disc and incisure structure. I have only a few comments mostly about presentation.

We are happy that the reviewer liked our study.

1. Because this study employs both frog and mouse, the authors should be careful to give the species when describing their results. The naming of the species would be particularly important in the first paragraph of the Results section and the legend to the first data figure, Figure 2.

We have clarified the text and figure legends to allow the reader to better follow which species each result came from.

2. It is unclear what Movie 1 adds to Figure 3. This movie could perhaps be omitted.

We prefer to include the source data that the image in Figure 3 is derived from, particularly to stress that the pattern shown in a single z-section in this figure can be seen throughout the entire tomogram. We hope that the reviewer would agree.

3. Movies 3 and 5 either don't work or consist of single frames, which would be better illustrated as figures in the text rather than as supplementary movies.

We appreciate the reviewer catching that there were some technical issues with video playback. We have recompressed these videos and ensured that they will now play appropriately across a wide variety of computer specifications and video player applications.

4. The incisures in Figure 4 will be difficult for many readers to visualize. My experience was that once I saw one of them, I began to see the others. The incisures in Figure 5 are, on the other hand, very easy to see. If Figure 5 had come before Figure 4, I would have had no problem. The authors may wish to exchange these two figures or to supply a cartoon for one of the rods in Figure 4, so that the reader can more easily understand what he or she should be trying to see.

We thank the reviewer for pointing out that some readers may have difficulties in fully appreciating the structure of incisures in this figure. We made two changes to improve the presentation of these images. First, we pseudo-colored several examples of enclosed discs in Figure 3, which highlights the structure of incisures. We also indicated one example of an incisure in these images with an arrowhead. Second, we pseudo-colored the example shown in Figure 4A to illustrate the same point, while still allowing the reader to view the three remaining examples in Figure 4 without any overlaid modifications.

5. It is unclear to me why the authors are so fond of their untested theory that incisures "likely serve to protect the flat lamellar disc membranes from undesirable deformations" but seem skeptical of the notion that incisures are present and especially numerous in rods of large diameter to aid longitudinal diffusion. The later notion is supported not only by theoretical calculations but also by common sense.

We appreciate this comment and, in fact, feel agnostic about both of these not mutually exclusive ideas. We removed the statement that the deposition of peripherin-2 in incisures likely serves to protect the flat lamellar disc membranes from undesirable deformations from the Introduction and rephrased the text in Discussion to stress that both functions are plausible and not mutually exclusive.

This manuscript presents a clear and convincing description of disc formation and the role of the protein peripherin in the formation of disc incisures.

Thank you for your kind comment.

Reviewer #2:Summary: The manuscript by Lewis et al. focuses on the potential mechanisms underlying formation of incisures in rod photoreceptors. Incisures refer to the indentations that occur on the rim of the photoreceptor disc membranes. The presence of incisures has been noted for decades and have been identified across a number of species. The role of incisures is not entirely clear and the mechanisms governing their formation have largely been inferred from early transmission electron microscopy studies 40-60 years ago. More recent ultrastructural studies of rod outer segment discs from mice carrying mutant alleles of rhodopsin or periperhin-2 described changes in the length or presence of incisures, suggesting that these proteins likely play a fundamental role in incisure formation in mouse. The authors take advantage of advances in electron tomography to provide unprecedented analyses of incisure formation, size, and structural complexity in stacked discs within mouse photoreceptors. They also use genetic models to explore how rhodopsin and peripherin-2 contribute to incisure formation and length. The authors find that new discs are highly irregular in shape and do not contain incisures during disc formation. Incisures are only formed are discs are enclosed. They find that the incisures in adjacent discs always align adjacent to the ciliary axoneme. Intriguingly, they find evidence of physical connections on opposing sides of the incisure. Critically, they find that elevated levels of peripherin-2 increase incisure size and complexity while low levels of peripherin-2 prevent incisure formation. In contrast, reduced molar ratios of rhodopsin lead to smaller disc surface area but increased incisure complexity. These results lead the authors to conclude that incisure formation is mechanistically linked to the relative molar ratio of peripherin-2 to rhodopsin and that rods make a slight excess of peripherin-2 in order to drive proper disc closure. The excess peripherin-2 within the disc rim forces formation of an incisure.

We are happy that the reviewer liked our study.

Major comments1. Line 145-146: the location of the incisure adjacent to the ciliary axoneme is an interesting observation indeed. As frogs have a number of incisures, is this a similar observation in species with multiple incisures or more exclusive to those species with a single incisure?

Indeed, we did observe that one of the many incisures in a frog disc is aligned with the ciliary axoneme. We have now included Supplementary Figure S2 to highlight this observation using an example of two adjacent cells.

2. While the presence of non-axonemal microtubules aligned with incisures in frog rods may provide an explanation for the number of incisures, the correlation with peripherin and rhodopsin content was lacking. In other words, do frog rods have considerably more peripherin-2 per disc than mouse rods?

This is a great question and one that we are interested in pursuing in the future. However, adapting the mass spectrometry-based protein quantification approach that we used to determine the absolute numbers of peripherin-2, ROM1 and rhodopsin molecules per disc was a significant undertaking that took several years (Skiba et al., PMID: 36711880). This approach is currently applicable to only a particular set of outer segment proteins in the mouse and cannot be automatically re-purposed to quantification of proteins in other species that have different amino acid sequences. Thus, designing, validating and employing this quantification protocol to all peripherin-2 and ROM1 isoforms along with rhodopsin in the frog would be a major undertaking that cannot be completed in the context of this revision. Nonetheless, we are very appreciative for the reviewer’s enthusiasm for this topic and plan to address this question in the future.

Minor comments1. The location of the incisures are difficult to see in Figure 4. The arrowhead is pointing to a very low contrast area of the disc and the thin incisure can be seen, but it's difficult. If it is possible to pseudocolor the image in some way to highlight the disc vs the extramembrane space, it would be helpful.

We thank the reviewer for noticing this issue, which was also commented by another reviewer. As described above, we pseudo-colored several examples in Figures 3 and 4.

2. Line 138-142: As with any descriptive narrative of cell structures, it is important to ensure the reader can fully understand and appreciate the interpretation of the authors. The shape of the newly forming discs can be difficult to appreciate in Figure 4. The authors are strongly encouraged to perhaps take 1-2 examples and provide a drawing or schematic of the image that can be more clearly annotated to assist readers in finding the outline of the discs and incisures.

We appreciate this point and pseudo-colored the surfaces of new forming discs in the examples shown in Figure 4A. We feel that pseudo-coloring helps the reader better visualize not only the structure of incisures, but also the irregular shape of the newly forming discs as in this specific example.

Overall, the paper is well-written and organized logically. The figures are generally easy to interpret although some additional annotations would help readers identify incisures in some low-contrast images (see comments). The authors utilized state-of-the-art electron tomographic data and mouse genetics to address a fundamental question. This will be of considerable interest to cell biologists in the vision community and likely to the broader cell biology community on how peripheral/rim proteins can shape membrane.

It is needless to say that we are very pleased by these comments and that the reviewer found our study to be of considerable interest to the broader cell biology community.

The authors provide a well-reasoned model for how incisures form in mouse rod photoreceptors: a relative excess of peripherin-2 drives incisure formation. This agrees with their mass-spectrometry data and molar ratios of peripherin-2 and rhodopsin. The main concern and outstanding question is whether these results are specific to mouse photoreceptors? The experiments in *Xenopus* were limited and only found that a CRISPR knockout of one peripherin-2 ortholog prevented incisure formation. While this result agreed with the general model and how molar ratios of peripherin-2 contribute, the knockout phenotypes are different than that of mouse. Some hypotheses are mentioned to explain this, but none were tested. The authors provide a model that agrees with mouse data, but is this generalizable? The model should permit several predictions for incisure formation beyond that of mouse rods. It would be most helpful to look in a species with multiple incisures and calculate the molar ratios of rhodopsin and peripherin-2. Do *Xenopus* require significantly more peripherin-2 to form multiple incisures? Alternatively, is it possible for the authors to mine publically available proteomics studies to assess rhodopsin and peripherin-2 content from other species (e.g. human, non-human primates, rats, etc…) and correlate to incisure number and/or length? The study overall is interesting and thought-provoking, but the overall impact would be greatly enhanced if additional evidence was provided that their model is broadly generalizable given the variety in incisure number and photoreceptor disc morphology (e.g. surface area, diameter) across species.

Related to a comment above, we are interested in pursuing each of these directions in frogs and other species in future studies, although it is not feasible to accomplish the required body of work in the context of manuscript revision. There are three other photoreceptor tetraspanins homologous to peripherin-2 that remain to be quantified and knocked out in frogs, alone and in combinations (the xrds36 and xrds35 isoforms and the peripherin-2-like protein). Testing the function of each of these in incisure formation would be an endeavor spanning several years of work. Additionally, there are no publicly available proteomic datasets that would contain information allowing an accurate quantification of rhodopsin and these homologous tetraspanins in other species. This question would require us to adapt our protein quantification approach, which would indeed be valuable, but would take significant time to complete.

Reviewer #3:Summary. This work explores the formation of incisures in rod outer-segment (OS) disks. The visual pigment rhodopsin is the major lamellar protein in rod OS disks, while peripherin is the major structural protein of the disk rim. The authors used wild-type, Rho+/- and Rds+/- mice to vary the ratio of rhodopsin to peripherin in vivo, and compared these ratios to incisure length and complexity in rod OS.Comments. This study presents several new findings. The authors convincingly show by EM tomography that incisures only form after each OS disk has reached maturity (fully separated from the plasma membrane). This new finding corrects an earlier published observation. Next, they examined disk morphology in Rds+/- heterozygous null-mutant mice and showed that an ~50% reduction in peripherin levels resulted in rod OS disks with no incisures. They performed a similar study on Rho+/- heterozygotes mutants. This time they observed that an ~50% reduction in rhodopsin levels resulted in OS disks with excessively long incisures. MS analysis of rod OS proteins and quantitative analysis of the EM images showed that incisure length varies with the ratio of peripherin to rhodopsin. They further showed that wild-type rods contain a small excess of peripherin over the amount required to form mature disks with normal incisures. Finally, the authors examined the effects of peripherin levels in rods from *Xenopus tropicalis*, an animal containing large OS disks with multiple incisures and three homologs of peripherin. They used gene editing to generate *Xenopus tropicalis* with a null mutation in the xrds35 gene, which is most like mammalian peripherin. OS disks from xrds35-/- frogs contained no incisures by EM tomography, further supporting their hypothesis.

Thank you for this nice summary of our study.

Another protein in the rims of rod OS disks is ABCA4, an ATP-driven flippase that translocates PE conjugated to retinaldehyde from the lumenal to cytoplasmic leaflets of the disk membrane. Retinaldehyde is a toxic photoproduct of rhodopsin bleaching. It has been suggested that the large number of incisures in frog disks is due to the larger diameter of frog versus mouse rod OS, and hence the greater number of rhodopsins per disk. This relationship is thought to ensure sufficient ABCA4 flippase activity to process the larger flux of retinaldehyde released by rhodopsin in these wide disks during light exposure, and possibly to minimize the diffusion distance of retinaldehyde from the disk lamella to the rim. The authors' findings seem in conflict with this explanation. They may wish to comment on this facet of their results.

We agree that it is possible for incisures to promote the encounter rate between retinaldehyde and ABCA4 in the disc. We do not find this idea to conflict with any of our interpretations; rather, this may be a complementary function of incisures. However, we failed to find any place in the literature where this hypothesis has been explicitly proposed and feel uneasy to present it as a new idea of our own in discussion. Fortunately, these reviews are public and truly interested readers could appreciate this idea. It is also worth noting that the correlation between incisure number and disc diameter is not perfect. For example, owl monkey discs appear to have a large number of incisures despite having a similar diameter to the mouse (Kroll and Machemer, PMID: 4970987).

Significance. The manuscript presents novel and significant findings into an important cell biological problem, development of the rod OS. It is clearly written and the data are of high quality. The manuscript should be of broad interest to cell biologists and vision scientists. It is strongly recommended for publication.

We are glad that the reviewer found our study to be of broad interest.